# Understanding a Player’s Decision-Making Process in Team Sports: A Systematic Review of Empirical Evidence

**DOI:** 10.3390/sports9050065

**Published:** 2021-05-17

**Authors:** Michael Ashford, Andrew Abraham, Jamie Poolton

**Affiliations:** 1Faculty of Health and Life Sciences, Coventry University, Coventry CV1 5FB, UK; 2Research Centre for Sport Coaching, Leeds Beckett University, Leeds LS6 3QT, UK; A.K.Abraham@leedsbeckett.ac.uk (A.A.); J.Poolton@leedsbeckett.ac.uk (J.P.)

**Keywords:** decision-making, perception, cognition, action, information processing, recognition primed decision-making, ecological dynamics

## Abstract

Three perspectives were taken to explain decision-making within team sports (information processing, recognition primed decision-making, and ecological dynamics perspectives), resulting in conceptual tension and practical confusion. The aim of this paper was to interrogate empirical evidence to (1) understand the process of decision-making within team sports and (2) capture the characteristics of decision-making expertise in a team sport context. Nine electronic databases (SPORTDiscus, PsycINFO, PsycArticles, PsycTests, PubMed, SAGE journals online, Web of Knowledge, Academic Search Complete, and Web of Science) were searched until the final return in March 2021. Fifty-three articles satisfied the inclusion criteria, were analysed thematically, and synthesised using a narrative approach. Findings indicate that the relative absence or presence of mental representation within the decision-making process depends on factors, including complexity, typicality, time available, and contextual priors available in the game situation. We recommend that future research integrate concepts and methodologies prevalent within each perspective to better understand decision-making within team sports before providing implications for practitioners.

## 1. Introduction

Post-match diagnosis of team performance will often find individual or collective decision-making to be the difference between a win, loss, or draw. The importance it holds leads practitioners to seek understanding of how best to develop expert decision makers. This is not easy as team sports are often seen as unpredictable [1] environments, which require players to respond effectively to uncertain situations that vary both in time and complexity [2]. The scrutiny on decision-making proficiency in applied practice has compelled research to better understand the decision-making process and the characteristics of expert performance [3]. As a result, three clear perspectives have emerged—information processing, ecological dynamics, and naturalistic decision-making—that are born from inherently different views of human behaviour. The crux of the debate typically revolves around a player’s access to memory representations in the decision-making process. From an information processing account, players are seen to make decisions through a process of selection from formalised responses that are stored in memory;

“*Perceptual-cognitive skill refers to the ability to identify and acquire environmental information for integration with existing knowledge such that appropriate responses can be selected and executed (Marteniuk, 1976)*”.[4] p. 457

In contrast, the ecological dynamics perspective proposes that decisions are made through online perceptual control where perception and action are coupled through the information available in the environment, absent of cognitive resources [5]. For example;

“*This theoretical rationale proposes that the most relevant informational constraints for decision-making and controlling action in dynamic environments such as a rugby match are those that emerge during on-going performer-environment interactions, not information from past experiences stored in the brain*”.[6] p. 131

Unfortunately, the differences between these perspectives present knowledge-hungry practitioners with a juxtaposition of theoretical concepts, terminology, and practical implications that they may conflate or misinterpret in their design of learning activities and adopted coach behaviours. Seeing as each perspective disputes the way decision-making is understood, it is useful to present each view clearly. Those schooled in information processing have tended to adopt an expert performance approach [7] to understand the mechanisms and processes that underpin elite performance and discriminate elite players from their less skilled counterparts [8,9,10]. The resulting evidence presents decision-making as a deliberate process of selection, in which expert players excel in their capability to; extract and process cues from the environment [11,12] recognise and interpret familiar patterns of play [12,13,14] form expectations by computing situational probability [15,16,17]. These processes of selection are viewed as an intermediate agent between what a player perceives (perception) and how a player responds to the play unfolding about them.

In contrast to the aforementioned process, the school of ecological dynamics describes how individual and team/shared behaviour emerges as a result of; an ongoing reciprocal relationship between perception of information, which constrains movement, and action, which creates information [9,18,19,20]. The coupling of perception and action can be captured as invitations for action (or options), known as affordances [21], which are defined relative to the task goal, structure of the environment and the action capabilities of the performer. Whilst shared affordances, capture a collective perception of what is possible within the constraints of a context. Decisions of one player interacting with another (to give or receive a pass for example) are made based on the affordances offered by the environment and the perception of the capability of teammates who they are in a position to interact with.

Finally, naturalistic decision-making was conceptualised by Klein, Calderwood, and Clinton-Cirrocco [22,23] to explain human performance in highly pressurised, time constrained situations. They argued that decisions are made through a recognition primed process that alters from situation to situation according to the decision maker’s familiarity with the perceptual information available (visual, auditory, olfactory, etc.) and their context relevant knowledge base [22,23]. In this theory the decision-making process happens through one of three levels [22,23,24]. Simple match—represents a case in which the decision maker recognises a situation as typical as the goals, cues, expectations, and action response present themselves in an obvious fashion. Diagnose the situation—where the information is not provided in a typical fashion. Using a process of rapid story telling through mental simulation, the decision maker has to clarify the goals, cues, and expectations through a process of diagnosis to restore typicality and come to a decision. Finally, evaluate a course of action—where the information available (goals, cues and expectations) is recognised but a course of action does not immediately present itself. As such a course of action is rapidly mentally simulated considering intended and unintended consequences to disregard or select an appropriate course of action. We recognise that both the information processing and naturalistic decision-making perspectives are both grounded in a cognitive view of the world. However, we propose they are sufficiently different that they are worthy of being considered as separate approaches to examining decision-making in teams sports. They will form two separate perspectives throughout this study.

The presence of the three theoretical perspectives and their associated narratives presents problems for coaches attempting to use theory to inform their practice. First, researchers are often guilty of taking a firm theoretical stance and presenting just one side of the argument when making sense of the coaching problem and when interpreting their findings. Second, differences in the lexicon of the different perspectives can hamper relational and abstract thinking by coaches. Third, sharing of the findings via social media can result in nuanced misinterpretations of empirical evidence [25]. Fourth, national governing bodies of team sports may prescribe to one point of view and nurture a bias towards one way of looking at a coaching problem. As a consequence, a practitioner’s engagement with research can result in bewilderment, conceptual blind spots and convoluted solutions to an intricate practical problem. In an attempt to provide clarity, a systematic review of empirical studies on decision-making in team sports was conducted to (i) fully understand the process of decision-making within team sports and (ii) comprehensively capture the characteristics of decision-making expertise within a team sport context.

## 2. Materials and Methods

### 2.1. Development of a Search Strategy

To start this review, a list of keywords was created by deconstructing the research aims [26]. These keywords were used to conduct a preliminary search on the SPORTDiscus database. The returns from the preliminary search were sampled to identify the relevance to the research aims (i.e., every 10th return) and mined to identify other possible keywords [27]. This process was repeated until the search typically returned highly relevant studies. The keywords originally entered also returned studies that were unrelated to the research aims, such as those associated with sport injury or decision-making in sports marketing. Subsequently, these terms were added to the search phrase using the NOT operator. The final search phrase emerging from this process was:
“Decision Mak*”ANDSport*AND(Expertise* OR Process* OR Coach* OR Tactic* OR Team Game* OR Percept* OR Action* OR Anticipat* OR Cue OR Knowledg* OR Affordance* OR Cognit*)NOT(Sport Injur* OR Adventure OR Recruit)

The following databases were accessed on the basis of relevance to the research question and accessibility to the lead researcher: SPORTDiscus, PsycINFO, PsycTests, PubMed, SAGE Journals Online, Web of Knowledge, Academic Search Complete, and Web of Science.

### 2.2. Inclusion/Exclusion Criteria

Inclusion/exclusion criteria were used to set clear boundaries to the review process and ensure returns would pinpoint all studies that met the research aims [28,29]. The studies included needed to: (i) be peer-reviewed research studies; (ii) be published in the English language; (iii) be published before March 2021 (when the final search criteria were established); (iv) have collected original empirical evidence; (v) have reference to decision-making in the title or abstract; (vi) only involve the investigation in the context of team sports [29,30]; and (vii) include data that related directly to the aims of the study; for example, studies reporting findings related to the development of player decision-making, but neither the decision-making process nor characteristics of decision-making expertise were omitted.

### 2.3. Search Returns

The initial search phrase was completed on the 10 May 2017, which returned 524 peer reviewed articles following the removal of duplicates. In applying the inclusion criteria, a further 359 articles were excluded for exploring the decision-making of athletes within individual sports, leaving a total of 165. The full text of each was then assessed against the inclusion criteria and research aims, which resulted in 123 articles being excluded. Of these 124, 36 were conceptual or review articles and 88 were articles that produced empirical data, the majority of which were excluded due to focusing on the development of player decision-making. Initially, a total of 41 papers met the inclusion criteria. Following feedback from an external panel of experts within the subject area in March 2021, a follow up search was conducted on 6 March 2021, for appropriate articles published after the initial search. 418 peer reviewed articles were returned and mined leaving 19 appropriate articles following the removal of duplicates. Of the 19 articles, 12 were excluded, 4 were conceptual, whilst 8 fell outside the scope of the inclusion criteria as they focused on external factors influencing the decision-making process (i.e., fatigue or emotional intelligence). A further 5 articles were signposted by the panel for inclusion having not arisen from either search. In total, 13 articles were added to the initial 41, leaving a total of 53 research articles meeting the inclusion criteria (See Figure 1).

### 2.4. Data Synthesis

The final 53 articles were read multiple times in full to capture the focus of investigation, method, findings, inferences and implications of each study [29]. Following this, a two-stage thematic analysis was completed to identify consistent themes within the data [29,31]. First, deductive analysis was used to identify data that informed the research aims [26,28,29]. Second, each study was rigorously explored and classified according to the perspectives that shaped the theoretical focus of investigation, the study design and/or the interpretation of the findings. This exploration enabled comparisons, similarities, and differences to be drawn between and within each perspective. There was a mix of quantitative and qualitative data sets across the 53 studies; therefore, to find a “middle ground” [29], p. 809 a narrative approach to synthesis was adopted to integrate, interpret, and communicate the relevant findings [32,33].

### 2.5. Establishing Trustworthiness and Audit Trail

Across the articles returned, an equal balance of quantitative and qualitative study designs were identified, which led us to take numerous precautions to ensure the trustworthiness of the review [29]. Trustworthiness is a term frequently used in qualitative research to describe the validity and reliability of a study’s method and findings [29,34]. In order to establish such trustworthiness, an audit trail was kept of the initial keywords, search terms, repeated search phrases and the search returns. Furthermore, the audit trail kept note of studies that were excluded following the application of the inclusion/exclusion criteria [26]. The audit trail was continuously reviewed and verified by a group of academics who have conducted research in this area or had a research interest in the area of decision-making in team sports [29,34]. This included providing support to the lead researcher, acknowledging and challenging personal bias when interpreting findings, shaping and guiding the methodology of the review, and the guidance of shaping the conclusive interpretations of the data. Additionally, in an attempt to abide by Tracy’s ‘Big Tent’ criteria of ensuring quality in our approach [35,36] a panel of expert researchers within the subject area (*n* = 2) of decision-making in team sports, offered an external appraisal of the methods, approach, and returns taken following the initial search.

## 3. Results

### 3.1. General Results

The 53 articles included in this systematic review comprised a total population size of 2078 participants, made up of 1552 males, 427 females, and 99 participants whose sex was not declared. Moreover, 2021 were team sport players and 57 were coaches. Table 1 is a summary of the research perspective, the level of the sample, the population size of the sample, the sport, the method, what was measured and an indication of whether the article assessed choice, perception or choice and perception.

### 3.2. The Decision-Making Process 

Forty-one articles explored the process of decision-making within team sports explicitly in their studies. Table 2 summarises these findings into the three broad perspectives. In seeking to extract a definitive decision-making process from the literature, clear descriptions (if available) have been directly quoted from relevant articles. The thematic analysis has shaped three broad processes that align with the three perspectives: perceptual–cognitive expertise, perception-action coupling, and recognition primed decision-making, respectively.

#### 3.2.1. Information Processing

##### Perceptual–Cognitive Expertise

Sixteen studies presented the decision-making process as one encompassed by a player’s possession of specific key perceptual–cognitive skills [40,41,43,44,53,55,56,64,66,69,70,71,73,82,83,84] namely; the utilisation of domain knowledge in perceiving informational cues [44,55,66], the identification of global, salient and predictive cues [40,41,64,69,73,84], rapid retrieval of knowledge from memory representations [40,43,44,56,69], option generation [40,64,66,69,73], and the role of intuition in the form of the take the first heuristic [40,59,64,73,80]. A concurrent theme is the prevalence of representation as a connecting mechanism between what players see and how they act. McRobert et al. [71] found that skilled cricket batsmen fixated on predictive cues, which were processed through the retrieval of information and afforded anticipation of future outcomes. In two studies, Roca and colleagues [82,83] found that skilled players were better able to verbalise the retrieval process. They proposed that players use task specific memory representations that allow them to perceive the most relevant cues, retrieve the most suitable response and perform the most appropriate action. McPherson and Vickers [70] found that elite volleyball players update memory representations with knowledge of current event profiles (kinematic patterns, strengths, weaknesses, previous patterns of play) to inform future performance known as action-plan profiles. The sixteen studies on perceptual–cognitive expertise appear to agree that expert decision makers possess a larger ‘database’ of task specific information [40,43,44,56,69]. This ‘database’ of information is described as a catalyst for the retrieval of task specific mental representations that can be grown and refined to facilitate each stage of a perception-cognition-action process [40,71].

Four of these studies investigated expert team sport players decision-making, in the form of the intuitive take the first heuristic or option generation processes [40,64,73,80]. Klatt et al. [64] compared the decision-making accuracy of elite Brazilian and German senior academy football players through a video decision-making task. Intuition was measured by assessing the accuracy of participants’ first options, whilst creativity was measured in the number of appropriate options participants were able to generate. Basevitch et al. [40]; high skill vs. low skill and Musculus [73]; expert vs. near expert also employed video based decision-making tasks that assessed participants decision-making process in line with the “less is more” take-the first heuristic or “more-is-more” option generation processes whilst manipulating time available to make a decision. Participant responses in all three studies were valued against an expert coaching panel who indicated a rank of which options were best in each trial. Klatt et al. [64] found that Brazilian players were more accurate than German players in their decision-making as they generated a higher number of options, whilst also being more accurate in their first option. Basevitch et al. [40] reported that experts were more accurate than near experts in their decision response, their intuitive responses were more effective when less time was available, whilst generating more options was more effective when more time was available. Finally, Musculus et al. [73] only presenting findings supporting the less-is-more process, as participants who generated less options were more successful at indicating an accurate first response.

##### Dependence on Task Specific Declarative Knowledge

Six studies explored the role of consciousness within the cognitive control of the decision-making process [60,62,63,79,80]. Kinrade et al. [62] and Jackson et al. [60] both found that a player’s disposition to engage task specific declarative knowledge in decision-making (decision-reinvestment) or worry about the consequences of a decision (decision-rumination) predicted performance under pressure. Players with a raised tendency to ‘reinvest’ in task specific declarative knowledge or ruminate were more likely to suffer performance decrements when placed under pressure. Similarly, Raab and Laborde [80] found that players with a tendency to consciously process and deliberate over their decisions were less successful than those who acted through intuition.

Four studies considered the influence of the situation on the level of cognition employed by participants in the decision-making process. Four studies manipulated the complexity of information (number of attackers, defenders and options) and temporal constraints (time available) [40,63,73,79]. Kinrade et al. [63] found that tendencies for decision-reinvestment and decision-rumination were both negatively associated with performance on relatively complex tasks yet led to faster and more accurate decisions when the task was less complex. The amount of task specific declarative knowledge available to the players was not associated with the tendency to reinvest in this knowledge base under pressure, or to ruminate [63]. Raab [79] manipulated dependence on cognitive resources in learning either by providing a set of ‘if-then’ rules (i.e., task specific declarative knowledge) or by occupying cognitive resources with a subsidiary task (implicit learning condition). He found that task specific declarative knowledge facilitated transfer to relatively complex tasks, whereas, more implicit learning conditions led to superior performance when the task was less complex. Basevitch et al. [40] compared the anticipation, option generation and option prioritisation of high and low skill soccer players through a video-based decision-making task to explore automatic vs. analytical decision-making processes. Participants were required to watch video clips of footage from 11 vs. 11 game footage and once the screen occluded, anticipate what would happen, identify the possible options, and then prioritise those options by ranking them in their use. Additionally, the temporal constraints were time varied across three trials of 400 ms, 200 ms, and 0 ms and in cued (screen paused at point of occlusion) and non-cued (screen blacked out at point of occlusion) conditions. Skill based differences between groups indicated that high skilled participants automatic/intuitive and analytical decision-making complimented each other. More time gave higher skilled players an increased opportunity to explore and analyse all options successfully, whilst less time demanded a successful automatic/intuitive response. Musculus [73] employed a comparable video task with soccer players, with short-time (7.5 s) and long-time (30 s) of a paused frame at the point of occlusion. Their findings presented that intuitive decision-making was more effective for both near experts and experts across both conditions.

#### 3.2.2. Ecological Dynamics

##### Perception-Action Coupling

Eleven studies explored the decision-making process as a reciprocal relationship between the player’s perception of the environment and the actions of the player [45,46,47,48,49,50,51,52,74,75,76]. Eight of the eleven studies analysed patterns in players movements in the context of the task environment [45,46,48,49,50,51,52,74]. Passos et al. [74] found that the distance of the defender from the touchline within a 2 (attackers) vs. 1 (defender) situation in rugby union influenced the attackers decision to pass, which was taken as evidence for decision-making as an emergent process constrained by the player’s capability. Similar inferences emerged from an evaluation of movement responses in relation to variables, including distances between attackers and defenders and the time it took to close tau [46], the posture of the players [52], physical height [45], and the manipulation of instructional and task constraints [45,49,50]. Across these six studies, the authors encapsulate this relationship under the term ‘affordances’, which are defined as invitations for action. Similar findings were presented by Correia et al. [48] and Correia et al. [50]. Correia et al. [48] found that skilled rugby player’s decisions were dependent on the emergence of gaps between defenders. The skilled group were found to make a decision to run or pass depending on whether a clear gap emerged between the defenders, whereas the less skilled group were found to take the first gap frequently, regardless of a better option being available. Later, Correia et al. [50] indicated that decisions emerged depending on the interaction between attackers, defenders, and the touchline as findings demonstrated patterns in players decisions to use lateral movement towards or away from the side-line in 1 attacking vs. 1 defending rugby union player tasks. Esteves, di Oliveria, and Araujo [52] suggested that a superior capability to perceive affordances can be attributed to the attunement of the player to available perceptual information and better calibration between the information perceived and the capability of the player to meet their intended goal.

Paterson et al. [75] found that players were led by their intention within the task (to score) and their action capabilities (shooting ability, accuracy). Players tended to minimise the risk of shooting inaccurately by only selecting targets where their probability of scoring was high. Paterson et al. [75] suggested that a player’s skilled intention was directly guided by their capabilities to score, as a direct result of the decision-making process being ‘grounded in action’. Finally, Pepping, Heijmerikx, and De Poel [76] found that players decision-making behaviour adapted when shooting towards a target, passing to a teammate or passing over longer distances. They attributed these findings to a relationship between a player’s action capabilities, in this case mainly passing capability, and the opportunities for action that were presented to them. In summary, studies adopting an ecological dynamics perspective, report findings that were proposed to support the notion of decision-making as a coupled process of perception and action that cannot be separated.

##### Co-Adaption and Shared Affordances

Ten studies explored the process of synergies, co-adaption, and shared affordances within team decision-making behaviour. Silva et al. [85] found that national level rugby players covered a greater width of the pitch in attack and defence relative to their regional colleagues when the width of the pitch increased, suggesting a collective movement response to changes to task constraints. Collective movement was also identified when an attacking team had a numerical advantage [86,88] and when the number of football goals on the pitch increased [89]. This was attributed to the notion of co-adaption, where teams implicitly adapt their collective response to changes to the constraints within the performance environment [86]. Similarly, Correia et al. [47] explored territorial gain dynamics within sub-elite rugby union players. Their design analysed twenty-two attack vs. defence second phases of play within the opponents twenty-two metre line and measured distance gained by the attacking team. They found that functional groupings of attacking players termed synergies were a likely indicator of increase distance gained, findings also suggested that distance gained was a variable, which may have distinguished between successful and unsuccessful attacks. Ramos et al. [78] used an action research design to consider the development of team synchronisation, synergies and collective functionality in match play over the duration of a season. Their findings demonstrate that appropriate training environments that represent match demands and increase variability are likely to result in an increase in synchrony of counter attacking play.

Travassos et al. [87] found high variability in the interpersonal distances of attacking and defending players at the start of the passing task yet identified a convergence in movement at the point of pass initiation. They suggested that the convergence was driven by attacking players (both ball player and supporting teammates) perception of their opponent’s capability to intercept the pass. Correia et al. [50] also found that higher variability of displacement trajectories between attacking teammates and defensive opponents led attacking players to demonstrate positive decision-making behaviour rather than risk averse behaviour. Similarly, Passos et al. [74], found that the position of the supporting player and the distance of the defender from the touchline in a rugby 2 vs. 1 situation both influenced the timing of a pass, suggesting that a player making a decision takes into account the action capabilities of others. Four of these studies [47,78,85,86] explored the emergence of synergies in teams collective decision-making behaviour. Findings indicating co-adapted behaviour were also identified in a 1 attacker vs. 1 defender rugby union task [50].

#### 3.2.3. Recognition Primed Decision-Making

Four studies [61,67,68,72] investigate the process of recognition primed decision-making in team sports. The findings of each study fit Zsambok and Klein’s [24] model of recognition primed decision-making, which proposes that the process a player follows to decide on the best course of action depends on the demands of the situation [24,61,67,68]. Mulligan et al. [72] reported that expert ice hockey players identify patterns of play that encapsulate salient information in an attempt to assess the typicality of the situation. Johnston and Morrison [61] found that expert rugby league players also possess higher levels of pattern recognition and tend to make decisions with little conscious thought when a situation is typical. Taken together the two studies are consistent with Zsambok and Klein’s [24] view that a situation that is rapidly perceived as typical may only demand a simple match between the situation and a decision as the goals, cues and expectancies present themselves in a rapid and simple fashion (Level I). A situation that is initially perceived as atypical requires diagnosis to decide on an appropriate course of action (Level II), [24]. Macquet and Kragba [68] found that as the typicality of the situation decreased, there was a greater requirement for players to deliberately make sense of the unfolding situation and assess the risk associated with, in this instance, running a pre-planned play; players were aware of the risk of a decision having a negative outcome (e.g., loss of the point being played; see also); [61,67,68]. The final level of decision processing proposed by Zsambok and Klein’s [24] describes, a situation that is perceived as typical yet leads to mental rehearsal of multiple courses of action in order to evaluate which course is likely to result in the best possible outcome (Level 3). Both Macquet [68] and Macquet and Krabga [67] found that in team sports players rarely verbalised decision-making processes that were consistent with this level of decision. Two studies explored the importance of pattern matching, although not explicitly aligned to the RPD process. Both Poplu et al. [77] and Gorman et al. [58] found recurrent relationships between a player’s capability to match patterns and the accuracy of their decision.

Two studies explored the influence of contextual priors on the naturalistic decision-making process of team sport players [57,65]. Levi and Jackson [65] interviewed professional football players for their perspectives on the impact of context on the decision-making process. Inductive thematic analysis of interview data indicated that dynamic themes, such as personal performance, score status, momentum and external/coach instruction, and static themes, such as match importance, personal pressures and preparation, shape and influence the nature of the decision-making process. Their findings suggest that positive momentum can lead to situational favourableness where players feel that the game is going their way. Winning can result in increased confidence and a higher tendency to take risks, whilst losing can increase risk averse behaviour. Gredin et al. [57] examined the impact of players’ judgments on available explicit contextual priors and anticipation in 2 vs. 2 video-based soccer decision-making tasks. Judgement and anticipation were measured through accuracy scores and verbal reports of their decision-making process. The findings indicated that expert players use knowledge to recognise explicit contextual priors to inform their judgment and anticipation.

### 3.3. Characteristics of Decision-Making Expertise

A total of thirty-six studies identified characteristics of decision-making expertise. The results of the thematic analysis have been summarised in Table 3. Combined in the synthesis were a total of 21 key characteristics that fall under the three broader characteristics of perception, action capabilities and knowledge. We have deliberately chosen not to present these against an assumed theoretical perspective, as we are simply trying to capture the characteristics of decision-making expertise presented by empirical research.

#### 3.3.1. Perception

##### Cue Identification

Twelve studies explored the use of cues within player decision-making [40,54,57,59,61,64,65,69,70,71,73,84]. Johnston and Morrison [61] found that skilled players were able to cluster higher order cues together into one source of information (e.g., the width of a defensive line), whereas less skilled players more readily focus on discrete bits of information (e.g., gaps between individual defenders). ‘Cue clustering’ is presumed to allow numerous sources of information to be seen through one ‘global’ cue, thereby accelerating and optimising the decision-making process [61]. Similarly, McPherson and Vickers [70] concluded that expert volleyball players attended to rich chunks of information, as single visual search fixations were congruent with players’ verbalisations of cues and tactical information. These findings suggested that experts tend to perceive rich chunks of information that allow earlier opportunities to act.

Across the studies the findings were consistent in differentiating skill level based on perception of salient information to predict the outcome of an opponent’s action [54,59,61,71,84]. For instance, McRobert et al. [71] found that skilled cricket players focused on salient information that was proximal to the bowler, such as the bowler’s hand, to anticipate the conclusive location of the ball at the point it would reach the batsmen. Furthermore, within a simulated model of a basketball 1-on-1 situation, Fuji et al. [54] artificially changed the timing and location of a defenders front foot. The adjustment had a direct impact on the attackers drive direction and the performance outcome. Jackson et al. [59] found that high skilled rugby players were less susceptible to deceptive information than their less skilled counterparts; implying that skilled performers have learned to discriminate genuine visual information from deceptive information.

Studies included within this review that have explored experts decision-making processes through the take the first heuristic [40,64,73], option generation [40,44,64,73], and contextual priors [57,65,69] demonstrate clear findings that experts are able to perceive information of a global and salient nature earlier than less skilled players.

##### Visual Search

Nine papers used visual search tracking technology to attempt to gain insight into the mechanisms underpinning decision-making in team sport [37,38,41,43,66,70,71,82,83]. Four of these studies found that higher skilled players tended to make more visual fixations than lesser skilled players [38,43,71,82]. McRobert et al [71] attributed the higher number of fixations made by skilled cricket batsmen to their tendency to search for additional locations to identify the gaps between fielders. In direct contrast, Lex et al. [66] found that more experienced soccer players made less fixations than their less experienced counterparts when a 11 vs. 11 situation was presented. Roca et al. [83] reported a similar pattern of findings, when the situation contained only 1 or 2 opponents or teammates. However, unlike Lex et al. [66], when Roca et al. [83] presented a full 11 vs. 11 situation more experienced players made relatively more fixations [83]. The authors of these studies appear to agree that differences reflect the ability of skilled players to adapt their visual search behaviour to the changing demands of the task [66,71,83]. Two outstanding findings were presented by Afonso et al. [39], who found that players fixate for longer when they are in-situ compared to when they respond to screen based stimuli in the laboratory [39] and Bishop [41] found that players’ saccadic eye movements were faster left to right than right to left and provided habitual reading from left to right in western culture as the most likely explanation.

#### 3.3.2. Action Capabilities

Six studies [43,45,50,52,74,75] explored players action capabilities as a characteristic of decision-making expertise. Bruce et al. [43] found that the lesser-skilled netball players made decisions independent of their capability to perform the requisite skill. Likewise, Esteves et al. [52] found that the decision on which side to attack a defender in a 1-on-1 situation was independent of expertise, but was significantly influenced by defender posture (i.e., foot placement). In this case, the action capability of the defender to regain a position to defend the basket influenced the attacker’s decision. The novice attacker’s posture gave away information regarding their upcoming drive direction, while intermediate attackers were better able to hide this information. In contrast, both Passos et al. [74] and Correia et al. [50] found significant relationships between rugby players tendency to run and the distance of the defender from the touchline. Passos et al. [74] suggested that the ball carrier’s capability (speed, skill) to run through the gap between the defender and touchline directly influenced the action performed. Similarly, Paterson et al. [75] found a significant relationship between a football player’s ability to shoot and the challenge point of the target (target size and distance) they chose to shoot at, as better players chose more challenging targets. The authors suggested that the players’ selections were partially based on perceptions of their capability to meet the task demands. Similarly, Cordovil et al. [45] found that the height of basketball players was associated with inefficient movement paths towards the basket, resulting in an updated decision response.

#### 3.3.3. Knowledge

##### Task Specific Declarative Knowledge

Fourteen studies identified the role of task specific declarative knowledge in decision-making expertise [38,40,44,56,57,58,61,64,65,69,70,73,77]. Afonso et al. [38] found that highly skilled volleyball players tended to be able to verbalise a greater number of key pieces of game specific information than their lesser skilled colleagues. Afonso et al. [38] referred to this task specific knowledge as condition concepts, whereas studies reporting comparable findings refer to consciously accessible clusters of task and domain specific information mental representations, which are recalled from long term memory [58,61,71,77]. Studies exploring expertise differences present consistent findings where increased retrieval of memory representation is related to more accurate intuitive processes [64,73], option generative processes [40], and recognition of contextual priors [57,65,69]. McPherson and Vickers [70] found that elite participants are better able to verbalise game specific information following in-situ events. They suggested that the superior recall of game specific information is a result of stored responses in long term memory, which they referred to as mental representations. Additionally, Furley and Memmert [56] found that recreational basketball players with a low working memory capacity (i.e., more limited cognitive resources] were more susceptible to being distracted by secondary task-irrelevant information than players who have a high capacity, suggesting that the availability of cognitive resource influences a player’s capability to use task specific declarative knowledge. Finally, Klatt et al. [64] defined expert soccer players’ use of creativity as the ability to create novel and appropriate solutions to problems. In order to measure this they measured the quantity and effectiveness of option generation, which they linked to players knowledge of where to look and why.

##### Collective Knowledge

Seven studies [42,45,55,67,68,78,81] explored the concept of collective knowledge within teams. A consistent finding is that shared knowledge of tactical information across a team affords better decision-making. Richards et al. [81] reported that players adoption of a shared mental model of tactical understanding resulted in a game-to-game increase in effective decision-making and team performance. In a similar vein, findings were reported that signalled the importance to team performance of a ‘playbook’ or shared tactical understanding [67,68]. Similarly, Ramos et al. [78], albeit from an integrated ecological dynamics and constructivist approach, initiated an action research design to improve a volleyball teams collective synchronicity and decision-making behaviour over the duration of a season. Their findings demonstrated that explicit collective cue perception, shared tactical understanding, having shared strategic game plan, shared anticipation, and prioritisation of roles and responsibilities were at the heart of learning throughout the season. Macquet [68] refers to the use of teams having a shared understanding of pre-programmed tactics which support better execution of coordinated patterns of play in high-speed match specific situations. Bourbousson et al. [42] found a total of 47 knowledge elements were shared amongst a basketball team before the start of a competitive match. Following the game, player recollections showed that the collective knowledge pool of the team diminished throughout the duration of the match.

Cordovil et al. [45] found that tactical instruction had a significant influence on the actions and movements of players. The finding was interpreted as tactical instructions directly influencing the players’ goal directed intentions and the decision that emerges. Finally, Memmert and Furley [55] found evidence to suggest that specific tactical information provided by coaches can result in players missing important pieces of information.

## 4. Discussion 

### 4.1. The Decision-Making Process

The first aim of this review was to use empirical research to better understand the process of decision-making in team sports. Table 2 classifies studies by perspective and presents the representative descriptions of information processing, ecological dynamics, and naturalistic decision-making processes that were extracted from the papers.

Interrogation of the data has unearthed views about the decision-making process that are shared by the different perspectives. Foremost, is the idea that perception of salient information actuates the decision-making process [40,44,48,61,64,68,71,73,77]. Taking an ecological standpoint, Esteves et al. [52] suggested that skilled players are better able to identify opportunities for action afforded by the task environment, which is consistent with the proposal by those taking an information processing view that skilled players are better able to identify salient [70], predictive [71], global cues [61] within the context of their intended goal. The ‘hunt’ for affordances/salient information appears adaptive and dependent on task demands [38,51,66,83]. It is noteworthy that all the studies reviewed conflate perception of the environment with visual perception, ignoring the prevalence of auditory cues in team sport, e.g., teammates talking to each other. This seems to be a particularly interesting route for investigation. It is important to know more about how perception of information of this kind is integrated into the decision-making process.

How team sport players use, interpret or act upon perceptual information reflects the differences of the three perspectives. Advocates of ecological dynamics suggest that players have an inherent perception of what is technically and physical possible (action capabilities) in the context of the intended goal [45]. In contrast, those taking an information processing or naturalistic view argue that players develop task specific representations of how (procedural) and why (declarative) to respond in a certain way, which are retrievable from long-term working memory [40,44,53,64,70,73]. The pool of task specific declarative knowledge is said to be continually updated with experience of competitive situations (i.e., current event profiles; [70]) or through an improved tactical understanding presented by the coach [68]. From an ecological perspective, competitive experience enables a refinement of what the performance environment affords via attuning the player to its salient properties and calibrating the players action capabilities to the perceptual information unfolding before them [52]. Refined perceptual attunement offers an ecological explanation for Jackson et al.’s [59] finding that experts are able to see through the deceptive acts of their opponents. Noteworthy, is the interpretation of data by Cordovil et al. [45] who concludes that expert actions are a result of an interaction between task constraints and coach-led instructional constraints (i.e., tactics). Cordovil and colleagues align themselves to an ecological view yet acknowledge a place for the cognitive processing of task specific declarative knowledge (tactical instruction). Interpretations such as this highlight a significant tension between the perspectives that centres on the presence (information processing and naturalistic decision-making) and absence (ecological dynamics) of mental representation. Taken at face value, the relative quantity of empirical studies aligned to information processing, adds weight to the argument for the presence of memory representations in the decision-making process. However, the reduced quantity of papers aligned to ecological dynamics may be a function of its recent arrival on the team decision-making landscape having been built on previous theoretical perspectives of dynamical systems theory [90] and ecological psychology [91]. Furthermore, the uneven representation of perspectives in the included papers may highlight the empirical challenges imposed on research taking a more holistic view of a problem; therefore, it is important not to dismiss this work purely on the quantity of evidence. Tensions can perhaps be calmed by the naturalistic perspective.

From the evidence associated with naturalistic decision-making, there is a suggestion that certain situations demand a cognitive assessment of perceptual information, whereas some situations require little to none. Experts appear to amend their visual search strategies dependent on the type of situation faced and these shifts in visual attention appear to correlate with players’ verbalisations of their cognitive processes [66,70,83]. This view is supported by Raab [79] who identified that the complexity of a task (e.g., number of teammates and/or opponents) dictates the process underpinning a player’s decision-making. Implicit processes lend themselves to low complex environments whereas more explicit processes are more likely to be used in high complex environments [79]. Situational complexity may be defined by player perception of the typicality of a performance environment [67,68]. Familiar environments afford a simple match of a response to the play unfolding (Level 1 of the RPD process) [67,68] underpinned by implicit processes [79], whereas atypical environments evoke explicit diagnosis of the decision-making problem (Level 2 of the RPD process). Parallels can be drawn here to Klatt et al.’s [64] findings regarding the complimentary use of intuition and option generation by elite Brazilian football players, which suggested that successful use of creative and intuitive decision-making processes may dependent on the situation presented in the game [64,92,93]. Furthermore, Basevitch et al. [40] found that the successful use of an intuitive process or option generation was dependent on the temporal constraints (400, 200, or 0 ms) presented in the task. When expert players had more time, they generated more appropriate options, whilst when they had less time, intuitive processes were identified as being more accurate. Based on the evidence, it is logical to subscribe to the view that “cognition is best understood by looking at its environment” [93], cited in [79] p. 428.

The data has also suggested that the tendency to engage cognitive resources in decision-making is dependent on the player. Certain individuals depend more on conscious processes to select a course of action [60,63,80] particularly under pressure and sometimes inappropriately (e.g., low complexity tasks) [63]. In sum, the findings imply that the role of cognition in the decision-making process in team sports is fluid and dependent on the complexity/typicality of information available [79], the time available [40,67] and player disposition [60,62,63,80]. Furthermore, findings presented from Levi and Jackson [65], Gredin et al. [57], and Magnaguagno and Hossner’s [69] indicated that it is not only the complexity and temporal constraints of a situation that drive a decision-making process, but the explicit contextual priors that are available to a player. Dynamic contextual priors include assessment of personal performance, the score status, feelings of momentum and external coach/player instruction [57,65,69]. Whilst static contextual priors include the match importance, personal pressures, and a player’s preparation for a game [65]. Contextual priors capture a significant amount of social and psychological factors that can cause players to: be more confident, make risk averse decisions or risky decisions, reinvest in task specific declarative knowledge, feel pressured, experience feelings of situational favourableness, and identify strengths and weaknesses of teammates and opposition [57,65,67]. Levi and Jackson [65] and Gredin et al. [57] findings suggest that both dynamic and static contextual priors significantly influence the perception of game information and the decision itself [57,65,69]. Subsequently, this evidence suggests that a more integrated view of the decision-making process may be the best way to progress our understanding of a player’s decision-making in team sports.

#### Consideration of Methods

From the findings three distinct methods were unearthed, which included real time in-situ experiments (on field and lab based), a posteriori assessments of the decision-making process and verbalisation methods. Additionally, ten articles combined a posteriori evaluation with verbalisation methods of assessment [40,61,64,66,67,68,71,72,82,83] whilst two articles combined real time assessments with verbalisations [53,70]. Given the consistency in methods used across and between the fifty three articles it is essential to discuss (i) the study design taken; (ii) the measure/assessment of the decision-making process; (iii) when/how the decision-making process was analysed; and (iv) the consistency and validity of the method adopted. From the thematic analysis it is clear that the research perspective adopted by authors has driven the method taken to assess the decision-making process. Put simply, in the assessment of decision-making in team sports, paradigm seems to drive method.

Logically, the findings from each of these methods or combination of methods have resulted in contrasting findings regarding the decision-making process. Real time experiments of basketball players [52] and rugby union players [74] taken from the ecological view suggest that players actions and therefore their decision-making is dependent on the constraints of the task and their attunement to it. Despite these conclusions, no examination of perceptual attunement takes place. Instead, real time experiments assessing netball [43] and rugby union [59] from the information processing perspective indicate that previous declarative knowledge, procedural knowledge, technical ability, and perception of options all influence the resulting course of action [43,59]. It would seem that research has been somewhat constrained by a mixture of perspective driven methodology and a need to make a complex problem simple enough to research (see Table 1). The use of real time experiments without eye tracking, verbalisations, or interviews therefore results in findings that often overlook key elements of the decision-making process. Similarly, articles that have used a posteriori or verbalisation methods alone have also disregarded central components of the decision-making process explored elsewhere. For instance, Correia et al.’s [46,47,51] use of performance analysis measures of match footage assessed team decision-making to find that synergies and coadaptation account for collective decision-making in team sports. This approach relies on the validity of each researcher’s assessment of the game situation and a reliance on subjective inclusion of which game variables should form post hoc analysis. Furthermore, this body of work overlooks the importance of tactics and strategy, which are frequently highlighted as a key variable in successful decision-making in numerous articles included in this review [61,67,68,81,82,83]. Unsurprisingly, the same questions present themselves for articles that employ verbalisation/interview methods alone to assess the decision-making process, as task and environmental contexts are often ignored or over-exaggerated [16,65,84].

In contrast, more balance is found in the findings from studies that have employed a combination of methods, such as self-confrontation interviews [67,68,72], combination of video decision-making tasks with eye tracking measures [53,66,70,82,83], or combining video tasks alongside verbalisations [61,64,72]. The methods used within these articles allow for players perception, selection and a combination of both in conjunction with one another to be considered. Subsequently, the findings from these studies offer inferences that draw connections between deep declarative knowledge of their sport [61,64,66,67,68,71,72,82], their use of knowledge in their sport [53,70,82,83], the capacity to recognize [61,64,67,68] and make sense [61,64,66,67] of perceptual information offered within competitive situations and how these variable impact on the first options taken [61,64,68], and possible options that are available [64,68].

As a result of these findings there are key limitations in the methods adopted in the articles reviewed in this paper. Researchers drawing on ecological dynamics have focused on small decontextualized sub-phases of team sports, such as 2 vs. 1 situations in rugby union [74] or 1 vs. 1 situations in basketball [52], which, somewhat ironically, can lack representativeness because they do not fully capture variants in the complexity (typicality) of the criterion environment. For instance, Correia and colleagues [46,47,48,49,50] study of decision-making in rugby union considered a player’s decision to pass or run with the ball, but the authors offered generalised practical implications to coach all decision-making instances within rugby union. Given the findings regarding the impact of the game situation on the decision-making process presented in this review, the generalizability of practical implications offered by these authors should be questioned. Second, all five of Correia and colleagues papers [46,47,48,49,50] make the assumption that mental representations do not exist in the decision-making process leading them to ignore the question regarding the presence/absence of cognitive mechanisms in their method. We would argue that this approach demonstrates high levels of confirmation bias in reference to the research questions they pose. Consequently, future research investigating player decision-making in rugby union from the ecological dynamics perspective should employ methods that will test their conception of the decision-making process, not simply confirm it. In a similar fashion, research adopting information processing or naturalistic perspectives reduce the ecological validity through decoupling perception and action when using verbal [70] or non-representative action responses to video stimuli [71].

In light of the findings and limitations from each of the articles adopting real time, a posteriori and verbalisation methodologies, it seems imperative that future research ensures a full assessment of the decision-making process. That is, what a player perceives, the choices they have and how and why they come to their final decision. The findings from this systematic literature review suggest that future research attempting to understand and assess the decision-making process should combine the use of real time, a posteriori and verbalisation methods. Furthermore, the findings lead us to suggest that future research should consider an approach to study design and measurement that is not informed by a single perspective, which is, paradigm should not drive method. Researchers should consider manipulating or measuring the complexity (or typicality) of information through task (space, time, number of players), contextual priors (dynamic and static), or environmental (pressure, fatigue) constraints in order to explore how player decision-making may change. The measurement of such tasks might aim to capture specifying variables (visual, auditory, and kinaesthetic cues) [61,82,83], the presence/absence of cognitive mechanisms [64], and the resultant actions through retrospective performance analysis task analysis [68] or self-confrontation interviews [67]. A broader view on how the process of decision-making in team sports can be investigated and how data may be explained, may advance both our understanding of how players make decisions in team sports and the communication of findings to applied practitioners [94].

### 4.2. Characteristics of Decision-Making Expertise

The second aim of this systematic review was to comprehensively capture the characteristics of decision-making expertise within team sports. Higher-order characteristics of perception, action capabilities, and knowledge emerged from the inductive analysis of the findings (see Table 3) that each comprised of more specific characteristics of decision-making expertise.

#### 4.2.1. Perception

The identification of salient cues, predictive cues, and global cues within the performance environment have been presented as independent characteristics of perceptual expertise. Yet, closer inspection of the data suggest that, in a team sport context where temporal demands require players to anticipate the actions of opponents (and teammates), salient cues are most often predictive and are likely clustered into a global representation of the information [61]. Higher-order representations of salient information may underpin the concept of a simple match between perception and action [61,68] that allows experts to operate effectively under time pressure, as well as allowing experts to see through deceptive behaviour [59].

To provide insight into the information extracted by expert players, eye tracking technology has been employed. The research reviewed here identifies there is a level of ambiguity in the patterns of visual search data displayed by experts. Research suggests this ambiguity can be attributed to the specific demands of the task that attention is not necessarily aligned with gaze [66] or the decoupling of perception and action [95]. A cautious summary of the visual search data is that experts adapt their visual search behaviour according to the constraints of the task in order to extract salient information.

Despite no attempt to measure perceptual expertise (see Table 3), researchers adopting an ecological perspective in team sports have inferred that experts are better able to perceive opportunity for action (affordances) offered by the environment [48,74]. Support for such claims comes from research studying an individual sport. Berg and colleagues [96,97] compared experts and non-experts visual search during long jump performances. They presented evidence that the strategy of visual regulation of action in locomotion towards a target in space is not a function of extensive task-specific expertise, but instead the jumper can become better attuned to specifying information. Similarly, the superior decision-making behaviour of higher skilled players is assumed to be a consequence of the player being perceptually more attuned to the performance environment [52]. Further empirical work is needed to verify such claims in team sport environments.

#### 4.2.2. Action Capabilities

The concept of action capabilities is at the heart of ecological psychology and much of the data supports the notion that a player’s physical (e.g., speed) [74] and technical [75] attributes influence the action taken. Two studies failed to differentiate the decision-making of higher-level players from their lower-level and, presumably, less physically and technically capable counterparts. Bruce et al. [43] found that lesser skilled players made decisions that they were unlikely able to execute. Similarly, Esteves et al. [52] could not differentiate novice and intermediate attackers by their decision to attack the defender’s most advanced foot. However, their findings did show that novice attackers gave away postural information about their upcoming action, while intermediate attackers were better able to conceal this information. Esteves et al. [52] interpreted these findings as the novice players being perceptually attuned to the posture of their opponent before their action capabilities were calibrated sufficiently to successfully beat the defender. In other words, novice players could see the opportunity for action (i.e., the affordance) but could not accept the invitation. Advocates of ecological psychology contend that player’s need time interacting with the performance environment in order to recalibrate (or scale) their action (motor) system should their action capabilities have changed [98]. How much time is needed to calibrate effectively and whether experience moderates the time needed for and the precision of the calibration are pertinent questions for future research. Presumably, expert players are better equipped to deal with fluctuations in physical conditions across a match.

In summary, the notion that responses to an opponent’s action are subject to a player’s action capabilities suggest that experts physical and technical prowess, if calibrated, offers a wider array, and presumably more effective, opportunities for action (or tactical options). This idea closely parallels Launder’s pithy phrase [99];

“*what is tactically desirable must be technically possible.*”.[99] p. 59

#### 4.2.3. Declarative Knowledge 

The use of task specific declarative knowledge and the use of collective knowledge both emerged as characteristics of expertise (Table 3). There is a weight of evidence to suggest that experts possess a richer pool of task specific declarative knowledge [38,53,66,70,71]. Mental representations afford rapid selection of suitable action plans that allow experts to effectively operate in dynamic game environments [53]. Retrospective recall methods have been frequently used to gauge the quantity of task specific declarative knowledge accessible to players, but the methods are limited by assumptions that player’s accurately recall the knowledge used to formulate a response, e.g., [66]. Other ways of capturing knowledge have been employed and have tended to validate findings reported using recall methods, such as self-confrontation elicitation interviews [61] or the alignment of retrospective recall with visual search behaviour [70]. It is the job of research now to better understand how expert team players make best use of their more advanced declarative knowledge pool in competitive situations.

The data reviewed suggests that tactics are an extension of a player’s task specific declarative knowledge. Tactics, commonly imparted by a coach, guide players to key pieces of information [45,68], and allow them to respond to situations faster [66]. Decision-making expertise is not simply characterised by the knowledge of tactics, but how that tactical information can be operationalised in competitive game situations [66,81]. A caveat was put forward by Memmert and Furley [55] who argued that coach-led tactical instruction can blind a player’s perception of salient information. This assertion highlights the importance of coaches using tactics to scaffold the game for the players [2], while still allowing them to be attuned to salient information [55] and make use of individual players task-specific declarative knowledge [70] and experience [52] to identify opportunities for action. Indeed Pennington, Nicolich and Rahm [100] have long since suggested that allowing learners to elaborate procedural learning drawing on their own declarative knowledge, significantly supports transfer of that learning.

The use of tactical information has not only been explored at player level but also at team level. Shared mental models are presented as internalised tactical knowledge that extends to players having a shared view of salient information [81]. Teams who have a shared understanding of how they intend to play tend to be better able to coordinate more effectively and to make decisions in high pressure situations [67,68,84]. Shared mental models provide a framework for players to act within [2], but team adherence to the model can diminish across the course of a game [42]. This is may be a result of a team’s coordination through a shared mental model [81] being worked out by their opponent, which may demand a more emergent coordination of behaviour between teammates to achieve their intended goal [87]. Interestingly, Ramos et al.’s [78] study also employed an action research approach to improving team synchronisation in volleyball but from an ecological and constructivist perspective. Non-linear design principles are advocated as the central mechanism for the findings however the data clearly indicates that shared cue perception, shared tactical understanding, building a shared strategy and game plan, aiming for shared anticipation and shared priorities were all coach led practices used to improve the synchronicity of the teams counterattack behaviour. The similarities in method, approach, and strategies used throughout this action research seem to mirror that of Richards and colleagues [81] yet no reference is made to that work.

This explanation reflects conceptual research that has integrated the perspectives to explain team coordination [101]. Steiner et al. [101] consider shared mental models as a ‘top-down’ approach-in which internalised goal-directed tactics and behaviours drive coordinated group action-and the concept of shared affordances as a ‘bottom-up’ approach-in which group behaviour emerges from a shared inherent attunement to the opportunities for group action. Interestingly, these authors have stressed the importance of situational complexity on the nature of how decisions are executed, whether through shared mental models or shared affordances. In other words, team decision-making behaviour is dependent on the rules of the game and the demands of the situation, otherwise known as the internal logic of the sport. Steiner and colleagues [101] conception clearly implies that shared mental models [81] and shared affordances [74] sit at opposite ends of a team coordination continuum. Furthermore, co-adaption and synergies were highlighted a key discriminators of more successful decision-making teams, these papers have all identified collective patterns and functional movements of dyads (sub groups of players), but only through the assumption that this behaviour is emergent and self-organised [46,47,78,85,86,88]. It would be interesting to explore qualitative methods to explore the tactics and strategy that underpins team coordination, as it may unearth a more explicit motive behind the synchronicity and synergy demonstrated by players [101,102]. Thus, an appreciation of both ideas may help researchers and applied practitioners better understand how successful teams coordinate their actions and the key mechanisms behind it [94].

## 5. Conclusions

The interrogation of the empirical literature has identified a tension regarding the absence and presence of mental representations within the decision-making process that has been driven by differences in perspectives. However, an impartial appraisal of the data suggests that each perspective contributes to our collective understanding of decision-making in team sports. Decisions on how to act may be emergent [45,48,67], may be a product of a simple match [61,68,80] or require high-level diagnostic [68,82,83] or evaluative processing [4,67,68,81]. Therefore, the empirical evidence suggests that decisions can be placed on a continuum from bottom-up emergent behaviour to top-down evaluation according to the level of cognitive processing invested in the process. The early indication is that the complexity [79], typicality [68], time [40] and the contextual priors [57,65] within game information presented by dynamic team sport situations dictate the point a decision lies on the continuum.

The polarity of views is somewhat underscored by the lexicon adopted by different perspectives. Independent of perspective, perceptual expertise is defined by a player’s capability to identify the most salient information within the context of the intended goal [52,61] yet is described under three different terminologies, salient cues, perceptual attunement, and affordances. Long term working memory and perception of action capabilities describe factors that guide perception and are updated by current event profiles [70] and calibration [52], respectively. Decision-making processes are further influenced by tactics/coach-led instructional constraints [45,58] that reinforce goal-directed behaviour. At a team level, a group’s action is coordinated by reference to a shared mental model [80] and collective attunement to the shared affordances [6,87] offered by the game situation. Therefore, this systematic literature review formed the conceptual basis for Ashford, Abraham, and Poolton’s [94] communal language for decision-making in team sports.

It is our belief that the literature associated with understanding decision-making in team sports is selling itself short by failing to integrate ideas, accepting conceptual ideas over empirical evidence and accepting evidence and practical implications from unrelated sports and contexts that are fundamentally untested in the team sport domain. At present, research is this area is driven by specific perspectives that lead to interpretation of findings that fall victim to bias [103]. It may be better for empirical investigation of decision-making in team sports to be shaped by the rules of the sport [94] and the data examined through a variety of theoretical lenses to explore what happens, what works, and why.

## Figures and Tables

**Figure 1 sports-09-00065-f001:**
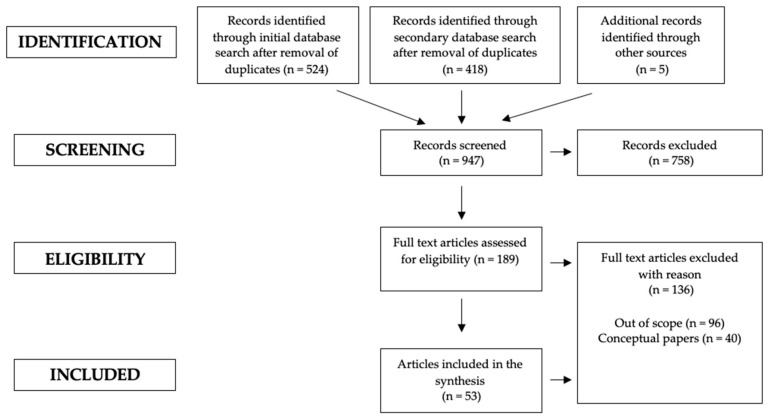
PRISMA flow diagram demonstrating the process of identification, screening, eligibility, and inclusion of research articles returned from the search phrase.

**Table 1 sports-09-00065-t001:** Summary of decision-making perspectives, level of sample, population size, team sport, method, measures and assessments of perception, selection or a combination used in the included articles.

Author(s)		Perspective	Level of Sample (as Defined in the Study)	Population Size	Team Sport	Method	What Was Measured	Assessment of Perception, Selection of a Combination?
**1**	Afonso [37]	Information processing	Elite	12 adult females	Volleyball	In-situ 6 vs. 6	Visual search and cognition	Combined
**2**	Afonso Garganta, McRobert, Williams, and Mesquita [38]	Information processing	Highly skilled and skilled	27 adult females	Volleyball	In-situ 6 vs. 6	Visual search and cognition	Combined
**3**	Afonso, Gargnata, McRobert, Williams, and Mesquita [39]	Information processing	Recreational	9 adult females	Volleyball	In-situ and lab based task	Visual search and cognition	Combined
**4**	Basevitch, Tenenbaum, Filho, Razon, Boiangin, and Ward [40]	Information processing	High skill vs. low skill	40 adult males	Soccer	Lab based video task	Anticipation and situational awareness	Selection
**5**	Bishop [41]	Information processing	Elite	13 adult females	Netball	Lab based task	Visual search and cognition	Combined
**6**	Bourbousson, Poizat, Saury, and Seve [42]	Ecological dynamics	Recreational	9 under 18 males	Basketball	Retrospective self-confrontation interview with real game footage	Shared knowledge	Combined
**7**	Bruce, Farrow, Raynor, and Mann [43]	Information processing	Expert, developmental and less skilled	58 adult females	Netball	Lab based and decontextualized in-situ task	Skill and decision-making ability	Combined
**8**	Causer and Ford [44]	Information processing	Skilled vs. less skilled	205 adults (females = 55, males = 155)	Soccer and invasion sports	Lab based video task	Situational awareness and decision accuracy	Selection
**9**	Cordovil, Araujo, Davids, Gouveia, Barreiros, Fernandes, and Serpa [45]	Ecological dynamics	Relatively experienced	10 adult females	Basketball	Decontextualized in-situ 1 vs. 1	Movement patterns	Selection
**10**	Correia, Araujo, Craig, and Passos [46]	Ecological dynamics	Semi-professional	Match footage	Rugby Union	Post hoc analysis of competitive fixtures	Tau (distance between attack and defence) and decision to pass	Selection
**11**	Correia, Araujo, Davids, Fernandes, and Fonseca [47]	Ecological dynamics	Semi-professional	Match footage	Rugby Union	Post hoc analysis of competitive fixtures	Movement patterns and territorial gain	Selection
**12**	Correia, Araujo, Cummins, and Craig [48]	Ecological dynamics	Non, recreational, intermediate, and professional	46 adult males	Rugby Union	Decontextualized virtual reality task-3 vs. 3	Decision-response	Selection
**13**	Correia, Araujo, Duarte, Travassos, Passos, and Davids [49]	Ecological dynamics	Youth recreational	12 males	Rugby Union	Decontextualized in-situ 1 vs. 2	Movement patterns	Selection
**14**	Correia, Passos, Araujo, Davids, Diniz, and Kelso [50]	Ecological dynamics	Youth recreational	8 males	Rugby Union	Decontextualised in-situ 1 vs. 1	Movement patterns	Selection
**15**	Correia, de Oliveira, Clavijo, da Silva, and Zalla [51]	Ecological dynamics	Experienced	32 adult males	Futsal	Post hoc analysis of In-situ futsal games.	Collective movement patterns and interpersonal distances	Combined
**16**	Esteves, de Oliveira, and Araujo [52]	Ecological dynamics	Youth novice and intermediate	32 males	Basketball	Decontextualized in-situ 1 vs. 1	Foot placement and movement patterns	Selection
**17**	Evans, Whipp, and Lay [53]	Information processing	Adult and youth-recreational	16 adult males	Football	Repeated in-situ 6 vs. 4 and verbalisations	Cognition, knowledge and pattern recognition	Combined
**18**	Fuji, Isaka, Kouzaki, and Yamamoto [54]	Information processing	N/A	Computer model	Basketball	Decontextualized simulated model-1 vs. 1	Anticipation	Selection
**19**	Memmert and Furley [55]	Information processing	Youth recreational	63 males	Handball	Decontextualized lab based tasks	In attentional blindness	Combined
**20**	Furley and Memmert [56]	Information processing	Recreational	69 basketball (36 male 33 female) and fifty five adult males	Basketball and ice hockey	Lab based and in-situ task	Decision-response and working memory capacity	Combined
**21**	Gredin, Broadbent, Williams, and Bishop [57]	Naturalistic	Expert soccer players	10 adult males and 8 adult females	Soccer	Lab based video task	Decision accuracy and judgment utility	Selection
**22**	Gorman, Abernethy, and Farrow [58]	Naturalistic	Expert and novice	32 adult males	Basketball	In-Situ 5 vs. 5	Pattern recall and decision-response	Combined
**23**	Jackson, Warren, and Abernethy [59]	Information processing	Skilled and novice	28 adult males	Rugby Union	Decontextualized lab based task 1 vs. 1	Anticipation	Combined
**24**	Jackson, Kinrade, Hicks, and Wills [60]	Information processing	County, regional and national	59 adult females	Hockey and Netball	Post hoc analysis of competitive fixtures	Cognition and decision-reinvestment	Selection
**25**	Johnston and Morrison [61]	Naturalistic	Professional and semi-professional	10 adult males	Rugby League	Retrospective verbalisations alongside game footage and game images	Cognition and recognition	Combined
**26**	Kinrade, Jackson, and Ashford [62]	Information processing	Recreational	111 participants (80 adult males, 31 adult females)	Basketball and Korfball	Comparison between coach classifications	Cognition and decision-reinvestment	Selection
**27**	Kinrade, Jackson, and Ashford [63]	Information processing	Skilled	38 adult males	Basketball	Low and high pressure lab based 2 vs. 2 and 4 vs. 4.	Cognition, decision response, decision accuracy and decision-reinvestment	Selection
**28**	Klatt, Noel, Musculus, Werner, Laborde, Lopes, Greco, Memmert, and Raab [64]	Information processing	Coaches Elite youth	62 adult male coaches and fifty six under 19 males	Football	Lab based video taskCoaches-Questionnaire	Creativity, intuition and cultural differences in decision-making.	Combined
**29**	Levi and Jackson [65]	Naturalistic	Professional	8 adult males	Football	Retrospective semi structured interviews	Situational factors influencing decision-making.	Combined
**30**	Lex, Essig, Knoblauch, and Schack [66]	Information processing	Less and more experienced	58 adult males	Football	Lab based retrospective verbalisations alongside game images	Cognition and decision-accuracy	Combined
**31**	Macquet and Kragba [67]	Naturalistic	Elite	7 adult females	Basketball	Retrospective self-confrontation interview	Cognition and decision-response	Combined
**32**	Macquet [68]	Naturalistic	Professional	7 adult males	Volleyball	Retrospective self-confrontation interview	Cognition and decision-response	Combined
**33**	Magnaguagno and Hossner [69]	Information processing	Expert vs. near expert	24 adult males	Handball	Lab based 1 vs. 1 task	Pattern detection, response and positional differences	Selection
**34**	McPherson and Vickers [70]	Information processing	Elite	5 adult males	Volleyball	Retrospective verbalisations following decontextualized in-situ 3 vs. 3	Visual search and cognition	Combined
**35**	McRobert, Ward, Eccles, and Williams [71]	Information processing	Less and more skilled	20 adult males	Cricket	Lab based task and retrospective verbalisations following real game footage	Visual Search and cognition	Combined
**36**	Mulligan, McCracken, and Hodges [72]	Naturalistic	Expert and non-expert	23 adult males	Ice-Hockey	Retrospective self-confrontation interview with real game footage	Familiarity and decision-accuracy	Combined
**37**	Musculus [73]	Information processing	Expert vs. near expert	169 adult males	Soccer	Lab based video task	Option generation	Combined
**38**	Passos Cordovil, Fernandes, and Barreiros [74]	Ecological dynamics	Recreational	24 males	Rugby Union	Decontextualized in-situ 2 vs. 1	Movement patterns	Selection
**39**	Paterson, Van der Kamp, Bressan, and Savelsburgh [75]	Ecological dynamics	Semi-professional	10 adult males	Football	In-situ and lab based free kick task	Decision-accuracy	Selection
**40**	Pepping, Heijmerikx, and De Poel [76]	Ecological dynamics	Recreational	8 adult males	Football	Decontextualized in-situ passing task	Physical and technical capabilities	Selection
**41**	Poplu, Ripoli, Mavromatis, and Baratgin [77]	Naturalistic	Expert and novice	48 adult males	Football	Lab based task alongside real game images	Decision-accuracy	Combined
**42**	Ramos, Coutinho, Ribeiro, Fernandes, Davids, and Mesquita [78]	Ecological dynamics	Youth recreational	15 females	Volleyball	Action research	Performance	Selection
**43**	Raab [79]	Information processing	Recreational	151 participants (26 adult female, 26 adult male, 99 unspecified)	Basketball and handball	Lab based tasks	Decision-accuracy	Combined
**44**	Raab and Laborde [80]	Information processing	Youth expert, near expert, and non-expert	54 (27 females and 27 males)	Handball	Lab based tasks	Decision-accuracy and decision-type	Combined
**45**	Richards, Collins, and Mascarenhas [81]	Naturalistic	Elite	1 female adult coach	Netball	Action Research	Performance	Selection
**46**	Roca, Ford, McRobert, and Williams [82]	Information processing	Skilled and less skilled	40 adult males	Football	Lab based task and retrospective verbalisations	Cognition and decision-accuracy	Combined
**47**	Roca, Ford, McRobert, and Williams [83]	Information processing	Skilled and less skilled	48 adult males	Football	Lab based task and retrospective verbalisations	Cognition and decision-accuracy	Combined
**48**	Schlappi-lienhard and Hossner [84]	Information processing	Elite	19 participants (11 adult females and 8 adult males)	Beach Volleyball	Retrospective semi structured Interviews	Decision-making characteristics	Combined
**49**	Silva, Travassos, Vilar, Aguiar, and Davids [85]	Ecological dynamics	Youth recreational	20 males	Football	In-situ small sided games	Collective movement patterns	Selection
**50**	Silva, Vilar, Davids, Araujo, and Garganta [86]	Ecological dynamics	Youth recreational	10 males	Football	In-situ small sided games	Collective movement patterns	Selection
**51**	Travassos, Araujo, Davids, Esteves, and Fernandesl [87]	Ecological dynamics	Intermediate	15 adult males	Futsal	In-situ small sided games	Collective movement patterns	Selection
**52**	Travassos, Goncalves, Marcelino, Monteiro, and Sampaio [88]	Ecological dynamics	Professional	12 adult males	Football	In-situ small sided games	Collective movement patterns	Selection
**53**	Travassos, Vilar, Araujo, and McGarry [89]	Ecological dynamics	Intermediate	15 adult males	Football	In-situ small sided games	Collective movement patterns	Selection

**Table 2 sports-09-00065-t002:** Summary of descriptions of the decision-making process used in the included articles.

Perspective	Articles Aligned	Clear Descriptions of the Decision-Making Process
**Information Processing** (*n* = 25)	Afonso [37] Afonso et al. [38,39] Basevitch et al. [40] Bishop [41] Bruce et al. [43] Causer and Ford [44] Fuji et al. [54] Jackson et al. [50] Jackson, Warren and Abernethy [59] Kinrade et al. [62] Kinrade, Jackson and Ashford [63] Klatt et al. [64] Memmert and Furley [55] Furley and Memmert [56] Lex et al. [66] Maqgnaguango and Hossner [69] McPherson and Vickers [70] McRobert et al. [71] Musculus [73] Raab [79] Raab and Laborde [80] Roca et al. [82,83] Schlappi-Lienhard and Hossner [84]	“Consequently, it appears vital that practice and instruction sessions include the coupling of perception, cognition and action components.” [71] p. 531 “These skills enable performers to make an assessment of the current situation and select appropriate decisions under time pressure.” [82] p. 301 “The first approach that explains intuitive and deliberative decision-making is an automatic information processing approach. It argues that intuitive choices are fast and subconscious associations between a perceived situation and a course of action… Intuitive decision-making from this perspective describes such choices as impulsive (Deutsch and Strack, 2008) or as “feeling is for doing” (Zeelenburg, Nelissen and Pieters, 2008). The main argument is that emotions can implicitly activate associated goals that manifest themselves behaviourally.” [80] pp. 89–90 “Decision-making is defined as the ability to use information from the current situation and the knowledge possessed about it so as to plan, select and execute an appropriate goal-directed action or set of actions.” [44] p. 1 “Conceptually, both constructs of intuitive and creative decision-making have in common that before a decision is made, option generation processes are involved, which bring about the options to choose from.” [64] p. 651 “To account for option generation and selection in sports, the theory of simple heuristics can serve as a theoretical starting point (Gigerenzer and Todd, 1999; Raab, 2012]. A simple heuristic is defined as a strategy that ignores part of the information, with the goal of making decisions more quickly, frugally and/or accurately than more complex methods.” [73] p. 272
**Ecological dynamics** (*n* = 18)	Bourbousson et al. [42] Cordovil et al. [45] Correia et al. [46,47,48,49,50,51] Esteves, de Oliveira and Araujo [52] Passos et al. [74] Paterson et al. [75] Pepping, Heijmerikx and De Poel [76] Ramos et al. [78] Silva et al. [85,86] Travassos et al. [87,88,89]	“There was some evidence to interpret decision-making as an emergent process under differing task constraints.” [45] p. 177 “As follows, decision-making in sport can be regarded as a goal-directed process of acting on the affordances available in the performance environment.” [49] p. 306 “…decision-making behaviours continually emerge from interactions between players and their surroundings. From this perspective, emergent decision-making behaviour has been conceptually defined as transitions in the action paths of performers.” [48] p. 244 “Decision-making and perception are both grounded in action that is, constrained by the action capabilities of the participants.” [75] p. 14 “When variability increases significantly, the system reaches a critical state of organisation, which prompts it to evolve. A region of self-organised criticality refers to a state reaches by a complex system near the border or edge of chaos.” [51] p. 297 “Decision-making can be regarded as emerging from constraints in the player-environment interaction that push the players to pick up informational variables about the possibilities for action afforded in the unfolding dynamics in order to accomplish performance goals.” [47] p. 985
**Naturalistic decision-making** (*n* = 10)	Evans, Whipp and Lay [53] Gorman, Abernethy and Farrow [58] Gredin et al. [57] Johnston and Morrison [61] Levi and Jackson [65] Macquet and Kragba [67] Macquet [68] Mulligan, McCracken and Hodges [72] Poplu et al. [77] Richards, Collins and Mascarenhas [81]	“…the NDM approach is also grounded in the premise that decisions are based on fast, pattern-matching processes which generally result in the rapid generation of one ‘sufficient’ option.” [72] p. 200 “…player’s decision-making was based both on a process of recognition of a typical situation and on the use of associations between a typical situation and a typical action. [68] p. 74 “It is widely recognized that decision-making depends on sense-making (e.g., Klein 2009). Sense-making is the process of analysing event retrospectively, explaining apparent anomalies, anticipating the future, and directing exploration of information.” [67] p. 346 “This model describes a decision process whereby a decision maker is first informed by pattern matching and informal reasoning, and options are compared to their compatibility with the situation, a process driven by the decision maker’s recognition of key features within the operational environment.” [61] p. 392

**Table 3 sports-09-00065-t003:** Thematic analysis-characteristics of decision-making expertise extracted from the articles.

Broad Characteristic	Key Characteristic	Article
Perception		
	**Cue identification**	
	Global cues	[40,44,57,61,64,65,69,70,73]
	Salient/Predictive cues	[40,54,57,59,61,64,65,69,70,71,73,84]
	**Visual Search**	
	Higher no. fixations	[38,43,71,82]
	Lower no. fixations	[66,83]
	Adaption of visual search behaviour to task	[39,41,66,71,83]
	Saccade latencies	[41]
Action Capabilities		
	**Action scaled**	
	Speed	[45,50,74]
	Skill	[43,45,74,75]
	**Body scaled**	
	Posture	[52]
	Height	[45]
Knowledge		
	**Task specific declarative knowledge**	
	Condition concepts	[38]
	Mental representations	[38,40,44,56,57,58,61,64,65,69,70,73,77]
	Working memory capacity	[56]
	Option generation	[40,57,65,69,77]
	Contextual priors	[57,65,69]
	Creativity	[64]
	**Collective Knowledge**	
	Tactical knowledge (shared mental model/playbook)	[42,45,55,67,68,78,81]
	Shared knowledge	[55,67,68,78,81]

## Data Availability

Data is contained within the article. The data presented in this study are available in the articles reviewed indluded in the reference list.

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
