# Peer review of "Understanding a Player’s Decision-Making Process in Team Sports: A Systematic Review of Empirical Evidence"

_sports, 2021, doi:10.3390/sports9050065_

Round 1
Reviewer 1 Report
In this well-written work, the authors aim to shed light on the decision-making process in team sports.
The method used in the research strategy is correct and carefully described. An important doubt remains on point VI of the inclusion and exclusion criteria. The Authors state: "only involve the investigation of team sports defined by the Oxford Dictionary of Sport Science and Medicine", but in many of the articles considered the paradigms used by the investigators did not meet this definition at all. The Authors do highlight this problem among the limitations of the study, but indeed the criticism persists. I will come back to this point later in my suggestions for Authors.
In this Systematic Literature Review work, the Authors raise a not trivial theme in research in the field of sports sciences, but their approach does not allow for clarity. The Authors persist, in my opinion, in the precisely error underlying this state of confusion. Instead of categorising the articles analysed by: consistency and validity of the observed variable with respect to the decision-making process (i.e., were the subjects really faced with a choice between two possible response behaviours?), paradigm used (i.e., 1vs1 vs multiplayers), how the decision-making process was assessed (i.e., evaluation by one/more experts vs. quantitative parameters), when the decision-making process was analysed (i.e., in real time vs. a posteriori by verbalization), sample size (i.e., does it respect statistical power), etc. The Authors chose to categorise the papers analysed according to the theoretical frameworks used to discuss the results of the articles. This leads to the persistence of the error which are intended to clarify. For example, if the task proposed in the study was to investigate the ability to collect/detect information (i.e., visual) can we talk about decision-making? Clearly not, no matter what theoretical framework the authors chose to discuss their results. Regardless of whether the authors refer to a theoretical framework that considers perception-action as indissolubly connected in a continuous loop, or as two separate factors of a computational process, the fundamental concept of a decision-making process (making a choice between alternative possibilities of behaviour in a context of uncertainty) is missing.
While appreciating the extensive work done by the Authors, I believe that at least four changes should be made to improve de paper impact:
1) Respect point VI of the inclusion/exclusion criteria. Therefore, exclude all work that does not meet the Oxford Dictionary of Sport Science and Medicine definition of Team Sport.
There may be a problem here as I could not find the definition of Team sport in the Dictionary and the reference of C. Swann et al. 2012 did not help. So I think the Authors were referring to the definition of Team:
“A social unit that has a relatively rigid structure, organization and communication pattern. The task of each member of a team is usually well defined, and the successful functioning of the team depends on the coordinated participation of all or several members of the team. Typically, the development of a team takes place in four stages: forming (members familiarize themselves with each other and start to form interpersonal relationships); storming (characterized by rebellion against the leader and interpersonal conflict); norming (during which hostility is replaced by cooperation with team members striving for economy of effort and task effectiveness); and performing (during which team members clarify roles and channel energies to achieve team success).”
According to this definition, all articles involving experimental procedures of less than 3 players vs. 3 players (both in ecological/naturalistic context or in video task) must be excluded from the systematic review.
2) Add to table 2.1 a column for: the population size who took part in the studies (specifying n° for each group).
3) Add to table 2.1 a column for: If Choice, Perception or Perception and Choice data were analysed in the article.
4) Add a paragraph, in discussion section, analysing the differences in the results found in the studies that proposed an evaluation design of the decision-making process in real time (i.e., during a game action), those that provided evaluation a posteriori (i.e., subjective evaluation by an expert) and those that use a verbalisation (i.e., retrospective interview).
Author Response
Reviewer 1:
In this well-written work, the authors aim to shed light on the decision-making process in team sports.
The method used in the research strategy is correct and carefully described. An important doubt remains on point VI of the inclusion and exclusion criteria. The Authors state: "only involve the investigation of team sports defined by the Oxford Dictionary of Sport Science and Medicine", but in many of the articles considered the paradigms used by the investigators did not meet this definition at all. The Authors do highlight this problem among the limitations of the study, but indeed the criticism persists. I will come back to this point later in my suggestions for Authors.
In this Systematic Literature Review work, the Authors raise a not trivial theme in research in the field of sports sciences, but their approach does not allow for clarity. The Authors persist, in my opinion, in the precisely error underlying this state of confusion. Instead of categorising the articles analysed by: consistency and validity of the observed variable with respect to the decision-making process (i.e., were the subjects really faced with a choice between two possible response behaviours?), paradigm used (i.e., 1vs1 vs multiplayers), how the decision-making process was assessed (i.e., evaluation by one/more experts vs. quantitative parameters), when the decision-making process was analysed (i.e., in real time vs. a posteriori by verbalization), sample size (i.e., does it respect statistical power), etc. The Authors chose to categorise the papers analysed according to the theoretical frameworks used to discuss the results of the articles. This leads to the persistence of the error which are intended to clarify. For example, if the task proposed in the study was to investigate the ability to collect/detect information (i.e., visual) can we talk about decision-making? Clearly not, no matter what theoretical framework the authors chose to discuss their results. Regardless of whether the authors refer to a theoretical framework that considers perception-action as indissolubly connected in a continuous loop, or as two separate factors of a computational process, the fundamental concept of a decision-making process (making a choice between alternative possibilities of behaviour in a context of uncertainty) is missing.
While appreciating the extensive work done by the Authors, I believe that at least four changes should be made to improve de paper impact:
1) Respect point VI of the inclusion/exclusion criteria. Therefore, exclude all work that does not meet the Oxford Dictionary of Sport Science and Medicine definition of Team Sport.
There may be a problem here as I could not find the definition of Team sport in the Dictionary and the reference of C. Swann et al. 2012 did not help. So I think the Authors were referring to the definition of Team:
“A social unit that has a relatively rigid structure, organization and communication pattern. The task of each member of a team is usually well defined, and the successful functioning of the team depends on the coordinated participation of all or several members of the team. Typically, the development of a team takes place in four stages: forming (members familiarize themselves with each other and start to form interpersonal relationships); storming (characterized by rebellion against the leader and interpersonal conflict); norming (during which hostility is replaced by cooperation with team members striving for economy of effort and task effectiveness); and performing (during which team members clarify roles and channel energies to achieve team success).”
According to this definition, all articles involving experimental procedures of less than 3 players vs. 3 players (both in ecological/naturalistic context or in video task) must be excluded from the systematic review.
Thank you for your comments. Our intention for this eligibility criteria was not made clear enough . We were guided by Swann et al. (2012) to state that the studies included needed to “(vi) only involve the investigation in the context of team sports (29, 30)” (page 4, lines 145-146). As a result, we have included Correia, Passos, Araujo, Davids, Diniz & Kelso’s paper, for example, which explores players choice of where to step and beat the defender in a 1 attacker vs 1 defender situation. This is a task in rugby union, with a rugby ball, that includes a decision. Therefore, it satisfies our inclusion criteria. Subsequently, in table 2.1. we have changed the subtitle from ‘Sport’ to ‘Team Sport.”
2) Add to table 2.1 a column for: the population size who took part in the studies (specifying n° for each group).
Thank you for your comment. This has now been included in table 1.
3) Add to table 2.1 a column for: If Choice, Perception or Perception and Choice data were analysed in the article.
Thank you for your comment. This has now been included in table 1 although we have used the language, an assessment of perception, selection or a combination.
4) Add a paragraph, in discussion section, analysing the differences in the results found in the studies that proposed an evaluation design of the decision-making process in real time (i.e., during a game action), those that provided evaluation a posteriori (i.e., subjective evaluation by an expert) and those that use a verbalisation (i.e., retrospective interview).
Thank you. We agree that the paper benefits from more direct consideration of the research methods employed to investigate decision making in team sports. In response, under the new sub-heading ‘Consideration of methods’ (page 11, 4.1.2.) we have added a full section in the discussion (Page 11-13, lines 702-791). It is clear to us from the thematic analysis that perspective (or paradigm) has driven the types of methods used by authors. Therefore, in keeping with the rest of the review, we have analysed the differences in methods used whilst also considering the role of research perspective.
Reviewer 2 Report
This manuscript addresses the factors that influence decision making in athletes. It reviews current and specialized literature.
The subject matter is interesting for the readers. The manuscript is adequately written and structures.
The methodology applied for the review seems appropriate.
I suggest to accept in present form
Author Response
This manuscript addresses the factors that influence decision making in athletes. It reviews current and specialized literature.
The subject matter is interesting for the readers. The manuscript is adequately written and structures.
The methodology applied for the review seems appropriate.
I suggest to accept in present form.
Thank you for your positive feedback.
Reviewer 3 Report
Dear Editor, the present review manuscript is focused on a interesting topic (the efficacy of decision making in team sports). It takes into consideration three perspectives to explain decision making within team sports (information processing, recognition primed decision making and ecological dynamics perspectives). The main outcomes are that the relative absence or presence of mental representation within the decision making process depends on complexity, typicality, time available and contextual priors available in the game situation. Recommendations are made for future studies to integrate concepts and methodologies prevalent within each perspective to improve decision making within team sports before providing implications for practitioners.
The ms presents a strong novelty and could add to the general level of knowledge on this topic. The methodological approach (literature research and study analysis) is robust and well-reported and cover all the steps required for this kind of article. The results and the relative discussion are adequately supported by the findings. I have just some minors the Authors should address.
1) Generally database such as PubMed and Scopus are taken in consideration during the study research phase. Why did not the Authors include also these database?
2) Page 4 lines 152-153: some more information about the reasons for excluding these articles are needed.
3) Please, check for some typos throughout the ms.
Author Response
Dear Editor, the present review manuscript is focused on a interesting topic (the efficacy of decision making in team sports). It takes into consideration three perspectives to explain decision making within team sports (information processing, recognition primed decision making and ecological dynamics perspectives). The main outcomes are that the relative absence or presence of mental representation within the decision making process depends on complexity, typicality, time available and contextual priors available in the game situation. Recommendations are made for future studies to integrate concepts and methodologies prevalent within each perspective to improve decision making within team sports before providing implications for practitioners.
The ms presents a strong novelty and could add to the general level of knowledge on this topic. The methodological approach (literature research and study analysis) is robust and well-reported and cover all the steps required for this kind of article. The results and the relative discussion are adequately supported by the findings. I have just some minors the Authors should address.
- Generally database such as PubMed and Scopus are taken in consideration during the study research phase. Why did not the Authors include also these database?
Thank you for your comment. PubMed was searched on both search phrases – we had simply made the error of not citing it in the manuscript. We have now included PubMed in both the abstract and the manuscript (Please see lines 13 and 135). Furthermore, Scopus was covered through the EBSCO databases that were searched which include all Elsevier journals (i.e. SPORTdiscus, PsychINFO, PsychTests, Web of knowledge, Academic Search Complete).
- Page 4 lines 152-153: some more information about the reasons for excluding these articles are needed.
Thank you. The papers that were excluded initially were those that explored the decision making of athletes competing in individual sports. Therefore, the following has been included in the manuscript on.
“In applying the inclusion criteria, a further 359 articles were excluded for exploring the decision making of athletes within individual sports, leaving a total of 165.” (page 4, lines 152-154)
- Please, check for some typos throughout the ms.
Thank you for the reminder to thorough proofread. As a result, we have rectified the typos that were identified.
Round 2
Reviewer 1 Report
I would like to thank the Authors for the changes made and the excellent revision of the article following my advice, I hope they will consider my comments as constructive to improve the impact of the article.